# PERSONALVIEW: MULTI-VIEW CONSISTENT HUMAN IMAGE CUSTOMIZATION VIA IN-CONTEXT LEARNING

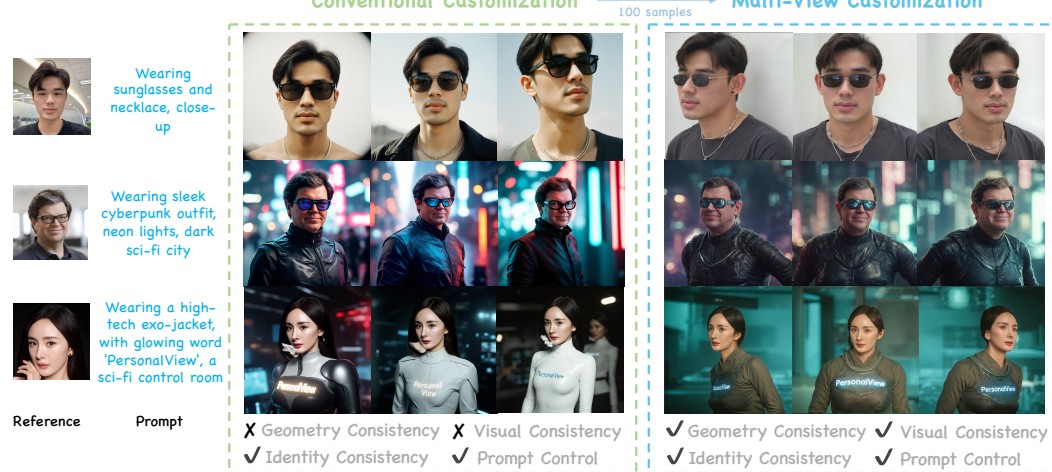

Figure 1: **PersonView compared to conventional customization methods.** PersonalView generates personalized images consistent with multiple views given one reference image. Conventional methods like PULID (Guo et al., 2024) have limited control over the viewpoint in the prompt (*i.e.*, left, middle, and right view) and do not have multi-view consistency.

## ABSTRACT

Recent advances in personalized generative models demonstrate impressive results in creating identity-consistent images of the same person under diverse settings. Yet, we note that most methods cannot control the viewpoint of the generated image, nor generate consistent multiple views of the person. To address this problem, we propose a lightweight adaptation method, PersonalView, capable of enabling an existing model to acquire multi-view generation capability with as few as 100 training samples. PersonalView consists of two key components: First, we design a conditioning architecture to take advantage of the in-context learning ability of the pre-trained diffusion transformer. Second, we preserve the original generative ability of the pretrained model with a new Semantic Correspondence Alignment Loss. We evaluate the multi-view consistency, text alignment, identity similarity, and visual quality of PersonalView and compare it to recent baselines with potential capability of multi-view customization. PersonalView significantly outperforms baselines trained on a large corpus of multi-view data with only 100 training samples. Generated samples are available at `https://personalview01.github.io/PersonalView/`.

## 1 INTRODUCTION

Recent years have witnessed an explosion of research endeavors in the domain of human image customization (Li et al., 2024e; Guo et al., 2024; Wang et al., 2024; Ye et al., 2023; Li et al., 2024c; 2025; Yang et al., 2024). These methodologies exhibit the capability for human customization to user-provided photographic input. For example, given a user's personal photograph, such approaches can synthesize novel customizations depicting the individual seated on a beach or smiling in a grassland setting. Despite the recent surge in human image customization techniques, a significant challenge persists in achieving multi-view consistent customization.

| Method | Geometry Consistency | Visual Consistency | w/o Large MV Dataset | w/o Test-Time Training | Human Identity Consistency | Prompt Control |
|---|---|---|---|---|---|---|
| Conventional Customization (*e.g.*, PuLID (Guo et al., 2024)) | ✗ | ✗ | ✓ | ✓ | ✓ | ✓ |
| Image to Multi-View (*e.g.*, Era3D (Li et al., 2024d)) | ✓ | ✗ | ✗ | ✓ | ✓ | ✗ |
| Unified Multi-Modal Model (*e.g.*, BAGEL (Deng et al., 2025)) | ✗ | ✗ | ✓ | ✓ | ✓ | ✓ |
| CustomDiffusion360 (Kumari et al., 2024) | ✗ | ✓ | ✓ | ✗ | ✗ | ✓ |
| PersonalView | ✓ | ✓ | ✓ | ✓ | ✓ | ✓ |

Table 1: **Comparison with previous methods.** In comparison, our method is capable of generating geometrically and visually consistent multi-view images while preserving human identity and prompt controllability, all without requiring large-scale multi-view datasets or test-time training.

What if a user wishes to change the view of the customized human while simultaneously synthesizing it in a novel context, as exemplified in Figure 1? Simply, employing viewpoint-specific prompts like 'from the left view' has limited control on customized images from diverse angles. Furthermore, the resulting outputs often suffer from a lack of inter-view coherence including *geometry consistency* of the human body and *visual consistency* manifesting across a range of visual attributes, encompassing body gestures, facial features, expressions, background elements, apparel, and accessories. Consequently, a salient inquiry emerges: how can multi-view consistency be preserved during the customized image generation process to satisfy user requirements?

In this work, we introduce a new task: Multi-View Customization (MVC) of human images conditioned on a single user-provided photograph. Without test-time tuning, it empowers users to generate multi-view personalized images exhibiting robust geometry consistency and visual consistency across different views with promising identity fidelity and versatile prompt-based control, as shown in Table 1. It not only enables users to explicitly control viewpoints for greater flexibility in customization and visual creation, but also holds potential for extension to other critical domains such as 3D modeling and reconstruction.

The core of this task is how to control the generation of multiple views while maintaining geometric and visual consistency. We are inspired by the in-context learning ability of the DiT-based models, such as FLUX (Labs, 2024b). As shown in prior work (Huang et al., 2024; Kang et al., 2025), they have the ability to generate a grid of images with roughly consistent visual content. This observation motivates our hypothesis that the model has a strong prior in multi-view capability, though the consistency is not guaranteed. Therefore, we propose to exploit this 3D-aware prior by in-context learning to foster more explicit multi-view consistency.

To this end, we propose employing in-context depth-maps as the cues to activate FLUX's capacity for geometry and visual consistent generation. Specifically, we introduce a novel generation framework based on multi-view depth-maps, termed **PersonalView**. At first, our goal is to obtain multi-view depth maps that are consistent with the provided prompts, which will serve as contextual cues for further processing. To achieve this, in the first stage, we leverage a pre-trained customization model (Guo et al., 2024) to perform preliminary sampling. Subsequently, we employ the SMPL model (Goel et al., 2023) for fitting and rendering, generating multi-view depth maps that align with the desired customization. Then, these multi-view human depth-maps are arranged into a four-panel grid, serving as an in-context conditioning signal for the depth-conditioned model to synthesize multi-view consistent customized images.

To enhance the geometry and visual consistency across multiple views, we further fine-tune the depth-conditioned adapter with a multi-view in-context learning paradigm. However, given the limited diversity of scenes within multi-view data, the model is susceptible to overfitting, potentially compromising its prompt controllability. To mitigate this, we introduce a Semantic Correspondence Alignment Loss specifically designed for DiT-based models to preserve the original model's semantic control capabilities. Concretely, this loss incorporates a frozen branch of the original model during training. By aligning the correspondence between text tokens and visual tokens extracted from both

the trainable and frozen branches, it effectively retains the original model's semantic control without impeding its in-context learning abilities.

Our proposed PersonalView consistently preserves multi-view human identity while demonstrating strong adherence to editing prompts, requiring only a few training steps without large-scale multi-view human dataset. It exhibits robust multi-view consistency across various attributes, including facial features, body gestures, background, and clothing, while concurrently preserving excellent prompt controllability and identity fidelity. Extensive experimentation demonstrates its superior performance compared to existing novel view synthesis methods that typically necessitate extensive training on large-scale multi-view datasets.

In summary, our contributions are as follows.

- We introduce a novel task, Multi-View Customization (MVC) conditioned on a single user-provided photograph without test-time tuning, which aims to generate multi-view consistent customized images with precise identity and diverse prompt-control.

- We propose PersonalView, leveraging in-context multiview depth maps to activate the in-context learning capabilities of DiT-based models with only a few training samples, facilitating the generation of geometry and visual consistent human images.

- We introduce a Semantic Correspondence Alignment loss tailored for DiT-based models, which preserves the original model's semantic control capabilities without impeding its in-context learning abilities.

## 2  RELATED WORKS

### 2.1  IMAGE CUSTOMIZATION

In the domain of Text-to-Image (T2I) generation, a variety of approaches have emerged to address identity (ID) customization (Gal et al., 2022; Li et al., 2024e; Gal et al., 2024; Valevski et al., 2023; Xiao et al., 2024; Ma et al., 2024; Peng et al., 2024; Li et al., 2023; 2024a). A seminal method in this line of work is Textual Inversion (Gal et al., 2022), which encodes user-specific identity information into a dedicated token embedding while keeping the T2I model parameters fixed. To improve identity fidelity, In contrast, encoder-based paradigms aim to directly inject identity representations into the generation pipeline. For instance, PhotoMaker (Li et al., 2024e) leverages large-scale identity datasets to construct robust ID embeddings from diverse image samples. Similarly, PuLID (Guo et al., 2024) introduces a more precise ID supervision mechanism by minimizing identity loss between synthesized outputs and reference images. Recently, CustomDiffusion360 enables explicit control over object viewpoints in the customization of text-to-image diffusion models; however, it requires test-time training and is limited to object-level customization rather than human subjects.

### 2.2  MULTI-VIEW IMAGES SYNTHESIS

Cross-view consistency plays a pivotal role in multi-view generation. MVFusion (Tang et al., 2023) initiates this direction by parallel multi-view image generation with correspondence-aware attention, which facilitates cross-view information exchange and supports textured scene mesh reconstruction. Building upon it, subsequent works (Tseng et al., 2023; Kant et al., 2024; Gu et al., 2024; Li et al., 2024b; Shen & Tang, 2024) incorporate epipolar constraints into diffusion models to enhance inter-view feature alignment. Zero123++ (Shi et al., 2023a) adopts a tiled view representation, enabling single-pass generation over multiple views, a strategy later adopted in Direct2.5 and Instant3D for efficient view synthesis. Similarly, MVDream (Shi et al., 2023b) and Wonder3D (Long et al., 2023) leverage dedicated multi-view self-attention mechanisms to promote cross-view coherence. Meanwhile, other approaches (Chen et al., 2024; Voleti et al., 2024; Yu et al., 2024) exploit spatio-temporal priors from video diffusion models to ensure view consistency across frames.

### 2.3  IN-CONTEXT LEARNING

Recent advancements in Text-to-Image (T2I) generation (OpenAI, 2023; Podell et al., 2023; Esser et al., 2024; Labs, 2024a) have enabled the synthesis of identity-preserving subject views within a

single $M \times N$ grid-structured mosaic, where each subview is prompted via carefully designed textual inputs. For example, IC-LoRA (Huang et al., 2024) explores in-context learning by fine-tuning a LoRA (Hu et al., 2021) model on concatenated grid-based image-prompt pairs; however, it suffers from reduced visual consistency, particularly in transferring identity across views. Similarly, by formulating the task as a grid-based image completion problem and replicating the subject image(s) within a structured mosaic layout, (Kang et al., 2025) demonstrates strong identity-preserving capabilities without the need for additional training data, fine-tuning, or modifications during inference.

## 3  METHOD

### 3.1  PRELIMINARY

**Diffusion Models.** Diffusion models (Sohl-Dickstein et al., 2015; Ho et al., 2020) are a class of likelihood-based generative models that produce data samples via a sequential denoising procedure originating from Gaussian white noise. During training, a predefined forward diffusion process is employed to transform clean observations $\mathbf{x}_0$ into a latent noise representation $\mathbf{x}_T \sim \mathcal{N}(\mathbf{0}, \mathbf{I})$ through the iterative injection of Gaussian perturbations across $T$ steps, forming a Markov chain structure, i.e., $\mathbf{x}_t = \sqrt{\alpha_t}\mathbf{x}_0 + \sqrt{1 - \alpha_t}\epsilon$. The model is trained to learn the backward process, i.e.,

$$p_\theta(\mathbf{x}_0|\mathbf{c}) = \int \Big[ p_\theta(\mathbf{x}_T) \prod p_\theta^t(\mathbf{x}_{t-1}|\mathbf{x}_t, \mathbf{c}) \Big] d\mathbf{x}_{1:T}, \tag{1}$$

Typically, the training objective maximizes the variational lower bound, which can be simplified to a simple reconstruction loss with the conditioning signal $\mathbf{c}$:

$$\mathcal{L}_{\text{diff}} = \mathbb{E}_{\mathbf{x}_t, t, \mathbf{c}, \epsilon \sim \mathcal{N}(\mathbf{0}, \mathbf{I})}[w_t || \epsilon - \epsilon_\theta(\mathbf{x}_t, t, \mathbf{c}) ||]. \tag{2}$$

**Diffusion Transformers.** A growing body of research has begun to adopt transformer (Peebles & Xie, 2023) architectures within text-to-image generative frameworks. Notably, models such as FLUX (Labs, 2024b) exemplify this trend by employing the MultiModal-Diffusion Transformer (MM-DiT) architecture. This design facilitates joint cross-modal interaction by performing attention over concatenated text and image embeddings, thereby enabling more effective integration of multimodal information during the generation process.

$$Q = [Q_T; Q_I], K = [K_T; K_I], V = [V_T; V_I], \tag{3}$$

$$\mathrm{A}(Q, K, V) = W(Q, K)V = \text{softmax}\left(\frac{QK^T}{\sqrt{d}}\right) V, \tag{4}$$

where $[;]$ is the concatenation, $Q$, $K$, and $V$ represent the key components of attention−query, key, and value, respectively; $Q_t$ and $Q_i$ correspond to the text and image query tokens; $W$ is the attention weight, and $A$ is the output correspondence.

### 3.2  TASK FORMULATION

Although existing human customization approaches have successfully demonstrated the ability to accurately preserve user identities and allow for various prompt-control, the generation of consistent multi-view customized images has yet to be thoroughly explored. In this work, we introduce a novel task: Multi-View Customization for human images. Given an image provided by the user and the prompt condition, our objective is to generate multiple customized images from different viewpoints that maintain high geometric and visual consistency across all perspectives, with robust identity similarity and prompt following.

### 3.3  IN-CONTEXT DEPTH-CONDITIONED GENERATION

Previous works (Huang et al., 2024; Kang et al., 2025) have demonstrated the contextual learning capabilities of the DiT model, leveraging this to generate images with consistent appearances. Inspired by these advancements, we hypothesize that the DiT model may also possess inherent multi-view

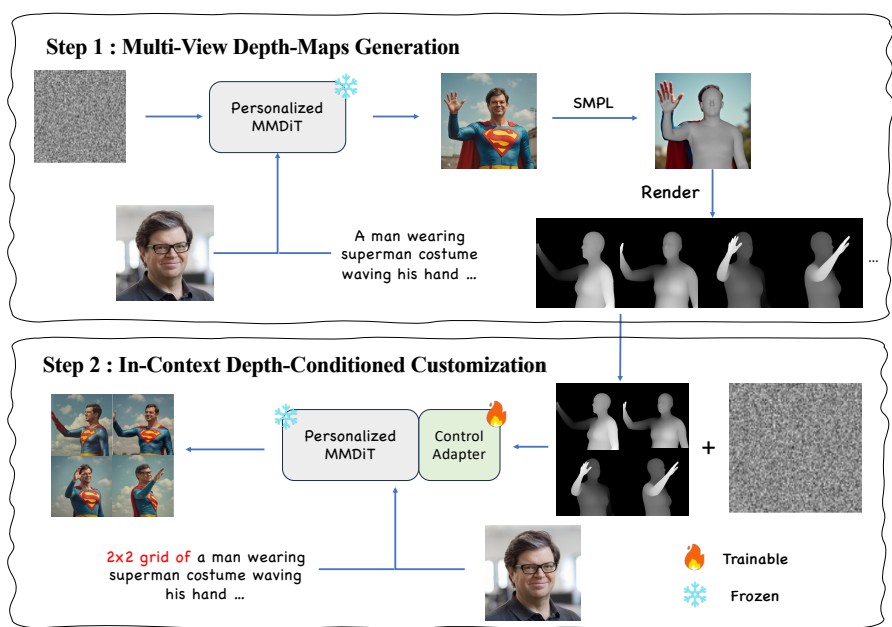

Figure 2: **Overview of PersonalView.** In step 1, we use SMPL (Goel et al., 2023) to fit the body mesh corresponding to the sample from the personalized generator (Guo et al., 2024). Then we render the body mesh for multi-view depth maps. With the in-context depth maps, we can generate the multi-view customization images in step 2 using the personalized model with our control Adapter.

geometric consistency. To explore this, we propose utilizing in-context depth maps as cues to activate the multi-view consistency of the DiT model.

Specifically, we adopt a two-stage approach as shown in Figure 2. In the first stage, our objective is to obtain multi-view depth maps that align with the provided prompt. Therefore, we leverage a pre-trained customization generator, like PuLID (Guo et al., 2024), to perform initial sampling based on the user-provided image and prompt. Subsequently, we employ SMPL (Goel et al., 2023) to fit a corresponding human body model and generate multi-view depth maps by rotating and rendering the fitted mesh. In the second stage, these multi-view human depth maps are arranged into a four-panel grid, which serves as an in-context conditioning signal for a pre-trained depth-conditioned model (Labs, 2024b). This setup enables the synthesis of multi-view consistent customized images.

### 3.4 LIGHTWEIGHT IN-CONTEXT LEARNING

With this in-context depth-conditioned framework, we can further fine-tune the depth-conditioned adapter with an in-context learning paradigm to enhance the geometry and visual consistency. Following FLUX, we adopt LoRA as the adapter mechanism. Empirical results demonstrate that, only a small amount of training samples is required to activate the model's inherent capability for multi-view consistency. Specifically, we sample multi-view human depth-image pairs from the multi-view dataset like NeRSemble (Kirschstein et al., 2023) and use the VLM model (Bai et al., 2023) to generate captions that correspond to the respective prompts. These depth maps and images are then organized into a four-panel grid format, which is utilized for in-context learning during training. To alleviate the adverse effects of background diversity, we further extract body masks $M$ from the depth maps and integrate them into the diffusion reconstruction loss as spatial priors,

$$\mathcal{L}_{\text{diff}} = \mathbb{E}_{\mathbf{x}_t, t, \mathbf{c}, \epsilon \sim \mathcal{N}(\mathbf{0}, \mathbf{I})}[w_t || \epsilon - \epsilon_\theta(\mathbf{x}_t, t, \mathbf{c})|| \cdot M]. \quad (5)$$

### 3.5 SEMANTIC CORRESPONDENCE ALIGNMENT

While the multi-view consistency is significantly enhanced during the contextual learning process, the relatively limited variety of scenarios in multi-view human data can lead to model overfitting, which in turn diminishes its ability to maintain semantic control. To address this degradation of semantic capabilities, we propose a Semantic Correspondence Alignment Loss as shown in Figure 3.

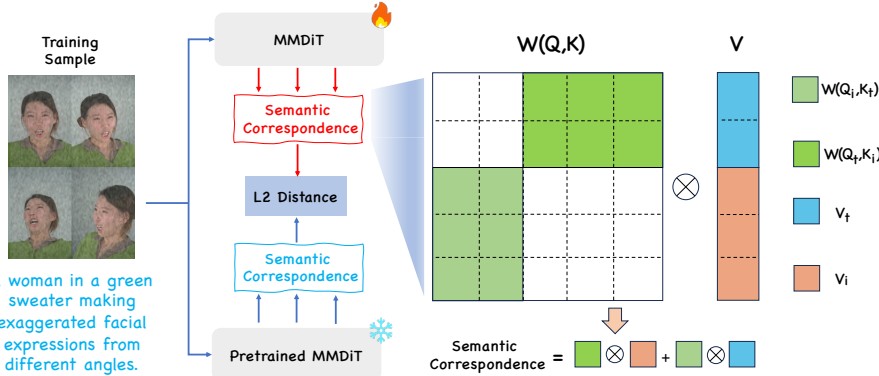

Figure 3: **Overview of Semantic Correspondence Alignment Loss.** Specifically, we minimize the L2 distance between semantic correspondences at each layer of the finetuned and pretrained MMDiT models for the same training sample, thereby explicitly constraining the finetuned model to retain the semantic control capabilities learned in pretraining.

Our primary goal is to preserve the model's original ability to respond to semantic textual inputs. To achieve this, we introduce a frozen branch of the pretrained model during training. Intuitively, aligning the correspondence between textual and visual tokens of the dual branch will facilitate the preservation of cross-modal semantic consistency without affecting its other behaviors. To the end, for the Query-Key-Value components in Equation (3) and Equation (4), we compute the semantic correspondence of $Q$, $K$ and $V$ in each layer $l$

$$SC(Q^l, K^l, V^l) = A(Q_I^l, K_T^l, V_T^l) + A(Q_T^l, K_I^l, V_I^l). \qquad (6)$$

Then we minimize the L2 distance between all the semantic correspondence of the fine-tuned and pretrained MMDiT

$$\mathcal{L}_{\text{SCA}} = \mathbb{E}_{\mathbf{x}_t, t, l} ||SC^F - SC^P||_2, \qquad (7)$$

where $SC^F$ and $SC^P$ are the semantic correspondence of the fine-tuned and pretrained MMDiT respectively. The alignment ensures that the model retains its semantic response capability while benefiting from multi-view consistency. Overall, our training loss is

$$\mathcal{L}_{\text{total}} = \mathcal{L}_{\text{diff}} + \mathcal{L}_{\text{SCA}}. \qquad (8)$$

## 4 EXPERIMENTS

### 4.1 EXPERIMENTAL SETUP

**Implementation details.** We utilize the recently developed DiT model FLUX (Labs, 2024b) with a pre-trained personalized module PuLID (Guo et al., 2024) as our foundational model. Our training set comprises only 100 cases randomly sampled from NeRSemble (Kirschstein et al., 2023), a widely utilized multi-view human dataset. Please refer to the Appendix for more details.

**Baselines.** To the best of our knowledge, our approach is the first to support end-to-end customized multi-view human image generation. For comparison, we construct a two-stage baseline wherein PuLID is used for identity-specific image synthesis, followed by the novel view synthesis method based on image-to-3D reconstruction. Specifically, we benchmark the proposed method against DiffPortrait3D (Gu et al., 2024) and Era3D (Li et al., 2024d), recent methods synthesizing 3D-consistent photo-realistic novel views. We also compare with ViewCrafter (Yu et al., 2024), a recent method synthesizing high-fidelity novel views of generic scenes. Besides, we compare with recent unified multi-modal model BAGEL (Deng et al., 2025) and Qwen-Image (Wu et al., 2025).

**Evaluation.** We collect a diverse portrait test set from the internet which consists of 80 characters ensuring demographic representation, with 50 prompts for comprehensive motion evaluation. To assess multi-view consistency, we follow the 3D consistency metric (MV Cons.) proposed in Pippo (Kant et al., 2025). MV Cons. is calculated on a pair of images where landmarks are estimated first and then reprojected to each other for error calculation. More details are in Appendix. The

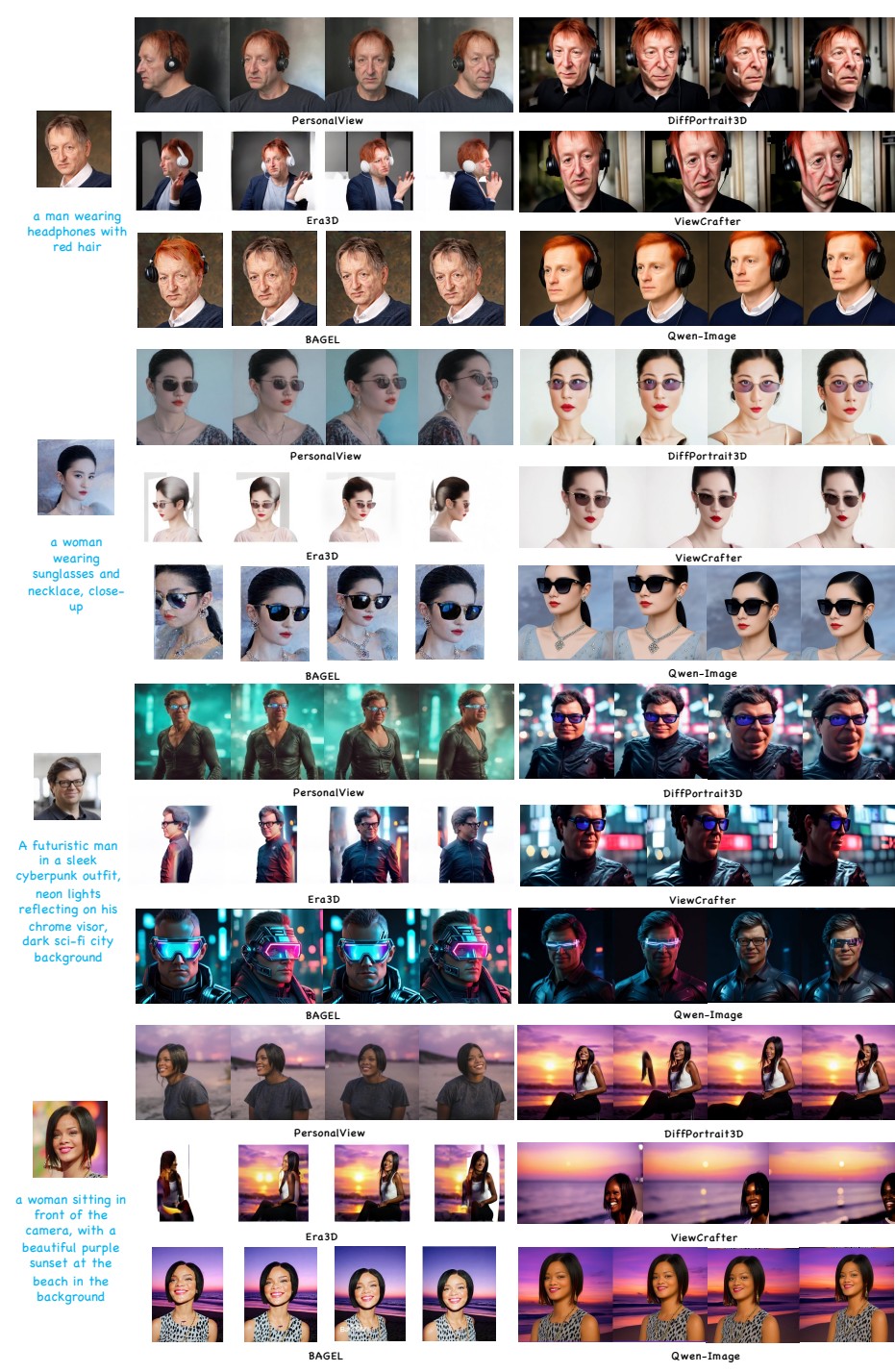

Figure 4: **Qualitative comparison.** DiffPortrait3D and Era3D exhibit limitations in maintaining geometric and visual consistency, especially with regard to full-body and background regions. Although ViewCrafter achieves improved scene modeling, it does so at the expense of geometric consistency in human representations. Besides, both BAGEL and Qwen-Image demonstrate suboptimal performance in terms of multi-view control. In contrast, our PersonalView achieves superior performance in both geometric fidelity and visual coherence across views.

evaluation framework employs CLIP-T and CLIP-I for quantitative assessment of text and image alignment. To further measure identity preservation, we implement facial recognition embedding similarity metrics (An et al., 2021) complemented by specialized facial motion analysis protocols.

| Method | MV Cons. (↓) | ID Cons. (↑) | CLIP-T (↑) | CLIP-I (↑) |
|---|---|---|---|---|
| DiffPortrait3D (Gu et al., 2024) | 7.887 | 0.7721 | 0.2588 | 0.6689 |
| Era3D (Li et al., 2024d) | 6.383 | 0.6462 | 0.2421 | 0.6708 |
| ViewCrafter (Yu et al., 2024) | 6.945 | 0.5820 | 0.2358 | 0.7037 |
| BAGEL (Deng et al., 2025) | 5.882 | 0.5623 | 0.2591 | 0.7289 |
| Qwen-image (Wu et al., 2025) | 5.324 | 0.6433 | 0.2603 | 0.7422 |
| **PersonalView** | **3.697** | **0.7920** | **0.2615** | **0.7793** |

Table 3: **Quantitative comparison.** We conduct a comprehensive comparison including the ability to achieve high multi-view consistency, identity consistency with the reference image (*i.e.*, ID Consistency and CLIP-I), and text alignment (*i.e.*, CLIP-T).

## 4.2 MAIN RESULTS

**Qualitative Results.** We present a qualitative assessment comparing PersonalView with baseline methods. As illustrated in Figure 4, DiffPortrait3D struggles with maintaining full-body geometric consistency and coherent scene appearance, largely due to its design and training focus on portrait-level generation. Similarly, Era3D, by primarily targeting subject-centric multi-view synthesis, performs poorly in preserving visual consistency across scene elements. Although ViewCrafter achieves improved scene coherence, it does so at the cost of human body geometry, often leading to distorted multi-view results. For BAGEL and Qwen-Image, they both exhibit relatively poor performance in multi-view control and consistency. In contrast, our proposed method achieves strong geometric consistency for the entire human body, along with visually consistent scene modeling, including accessories and backgrounds, while simultaneously preserving identity fidelity and supporting prompt-guided customization. More results are included in Figure 6 of the Appendix.

**Quantitative Results.** We report the results of the quantitative comparison in Table 3. As observed, DiffPortrait3D exhibits limited multi-view consistency, highlighting its weakness in modeling coherent geometry across viewpoints. Although Era3D mitigates this issue to some extent, it still suffers from subpar performance in identity preservation and semantic control accuracy. Likewise, ViewCrafter shows inferior results in both identity consistency and CLIP-based semantic alignment. Although BAGEL and Qwen-Image improve multi-view consistency, the identity similarity has decreased. In contrast, our method consistently achieves superior performance across all key evaluation dimensions, including multi-view consistency, identity fidelity, and prompt-aligned semantic controllability.

| Method | MV Cons. | Text Align. | ID Cons. | Overall |
|---|---|---|---|---|
| Era3D | 10.33 | 6.24 | 15.55 | 10.43 |
| DiffPortrait3D | 15.62 | 15.71 | 10.74 | 11.88 |
| ViewCrafter | 10.52 | 10.92 | 13.64 | 14.23 |
| BAGEL | 18.18 | 17.83 | 12.98 | 18.32 |
| Qwen-Image | 21.22 | 22.38 | 17.42 | 20.58 |
| **PersonalView** | **31.80** | **26.92** | **29.67** | **24.56** |

Table 2: **User Study.** Our PersonalView achieves the best human preference compared with all baselines.

**User Study.** To further evaluate the effectiveness of our methodology, we conduct a human-centric assessment, comparing our approach with existing novel view synthesis methods. We recruit 25 evaluators to assess 40 sets of generated results. For each set, we present reference images alongside multi-view images produced using identical seeds across various methods. The quality of the generated multi-view images is evaluated based on four criteria: Multi-View Consistency, Text Alignment, ID Consistency, and Overall Quality. As depicted in Table 2, our PersonalView achieves higher user preference across all evaluative dimensions, underscoring its superior effectiveness.

| Num. | MV Cons. (↓) | CLIP-T (↑) | ID Cons. (↑) |
|---|---|---|---|
| 200 | 5.553 | 0.2596 | 0.7902 |
| 400 | 4.282 | 0.2588 | 0.7899 |
| 800* | 3.694 | **0.2599** | **0.7912** |
| 1600 | **3.685** | 0.2579 | 0.7853 |

| Num. | MV Cons. (↓) | CLIP-T (↑) | ID Cons. (↑) |
|---|---|---|---|
| 100* | 3.697 | **0.2604** | **0.7920** |
| 200 | 3.752 | 0.2543 | 0.7899 |
| 400 | **3.688** | 0.2522 | 0.7882 |
| 800 | 3.694 | 0.2599 | 0.7912 |

(a) **Ablation study of training iterations.**      (b) **Ablation study of training samples.**

Table 5: **Ablation study for the number of training iterations and samples.** * indicates the best trade-off setting.

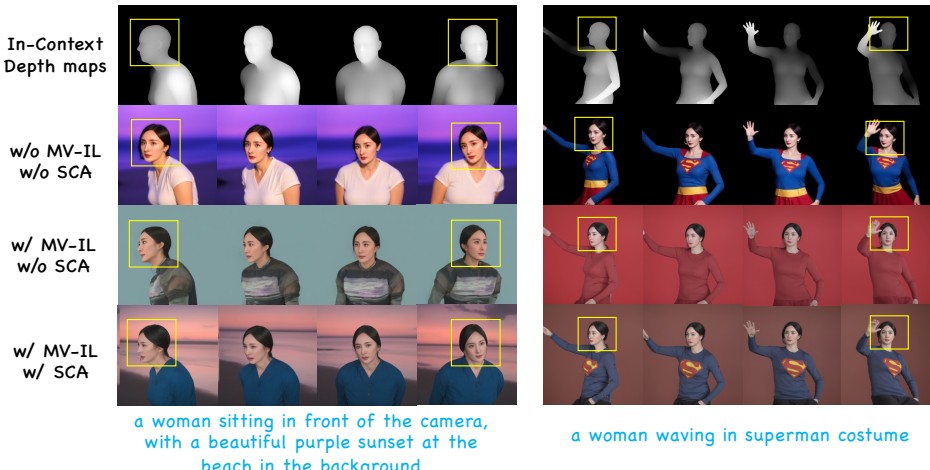

In-Context
Depth maps

w/o MV-IL
w/o SCA

w/ MV-IL
w/o SCA

w/ MV-IL
w/ SCA

a woman sitting in front of the camera,
with a beautiful purple sunset at the
beach in the background

a woman waving in superman costume

Figure 5: **Ablation study for multi-view in-context learning.** As shown, transitioning from the pretrained model (second row) to the multi-view in-context learning model (third row) significantly improves cross-view geometric consistency. The addition of the SCA module (fourth row) helps the finetuned model retain the pretrained model's ability to follow prompt-based semantic controls, such as identity, clothing, and background.

## 4.3 ABLATION STUDY

**Multi-View In-Context Learning.** To validate the effectiveness of our proposed components, we conduct an ablation study presented in Figure 5 and Table 4. As shown in the figure, the pretrained model often generates head orientations inconsistent with the corresponding depth maps, leading to degraded geometric consistency across viewpoints. Incorporating multi-view in-context learning significantly alleviates this issue, resulting in more coherent human geometry across views. However, due to the limited diversity of scenes in the multi-view dataset, prompt-based semantic controllability, particularly for attributes such as background and clothing, tends to diminish. In contrast, Semantic Correspondence Alignment (SCA) loss effectively preserves fine-grained semantic control, ensuring that identity and prompt-relevant attributes are maintained throughout the generated views. More ablation studies are given in the Appendix.

| MV-IL | SCA | MV Cons. (↓) | CLIP-T (↑) | ID Cons. (↑) |
|---|---|---|---|---|
| | | 7.270 | 0.2589 | **0.7943** |
| ✓ | | 3.928 | 0.2271 | 0.7753 |
| ✓ | ✓ | **3.697** | **0.2604** | 0.7920 |

Table 4: **Quantitative ablation study of multi-view in-context learning.** As observed in Figure 5, applying multi-view in-context learning improves multi-view consistency but degrades semantic controllability and identity consistency. Incorporating the SCA loss allows our model to recover these capabilities, balancing geometric consistency and semantic fidelity.

**Number of Training Iterations and Samples.** To investigate the impact of training iterations on model performance, we present a quantitative comparison in Table 5a. As indicated by the results, 800 iterations provide a favorable trade-off between efficiency and performance, and are therefore adopted in our final setting. Additionally, to validate the lightweight nature of our in-context learning framework, we evaluate the effect of varying the number of training samples. As shown in Table 5b, our method achieves competitive performance with as few as 100 training samples, demonstrating its data efficiency and strong generalization capability under limited supervision.

## 5 CONCLUSION

In this work, we introduce PersonalView, a novel framework for multi-view human image customization from a single photograph. By leveraging in-context learning with multi-view depth maps, our method enhances DiT-based models' multi-view reasoning. We also propose a Semantic Correspondence Alignment Loss to preserve prompt controllability during fine-tuning. Extensive experiments show that PersonalView excels in identity fidelity and coherent multi-view synthesis, requiring minimal additional training. Our approach enables efficient, high-quality multi-view customization without large-scale multi-view datasets.

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

# A APPENDIX

## A.1 IMPLEMENTATION DETAILS

For the NeRSemble dataset, we employ Qwen-VL (Bai et al., 2023) to generate descriptive captions. During training, we learn the LoRA for 800 iterations with a learning rate of 1e-4 with batch size 1. We employ the AdamW optimizer with a weight decay parameter of 1e-2. The epsilon is set to the default 1e-8 and the weight decay is set to 1e-2. During inference, we use 50 steps of DDIM sampler and classifier-free guidance with a scale of 7.5. We generate multi-view images with 512 × 512 spatial resolution. We used Lora rank 128, following the common setting of FLUX finetuning. All experiments are conducted on a single NVIDIA A800 GPU. Our code will be open-source.

## A.2 EVALUATION METRIC FOR MULTI-VIEW CONSISTENCY

To evaluate the multi-view consistency of the generated results, we adopt the re-projection error metric proposed in Pippo (Kant et al., 2025). Specifically, we first estimate facial landmarks from the generated images and then establish pairwise correspondences of these landmarks across different views. Based on these correspondences, we apply Triangulation using the Direct Linear Transformation algorithm to recover the 3D positions of each landmark. Finally, we reproject the 3D landmarks back onto each view and compute the Reprojection Error as the L2 distance between the original 2D landmark and the reprojected point, normalized by the image resolution. The final RE score is obtained by averaging this error across all views and landmarks.

## A.3 MORE ABLATION STUDY

The core design of SCA is to preserve the original model's semantic control (text-image correspondence) without compromising in-context learning during fine-tuning. In contrast, applying regularization on other aspects like full-attention features or model parameters leads to reduced multi-view consistency as shown in Table 7, since it disrupts in-context learning.

| Method | MV Cons. (↓) | CLIP-T (↑) | ID Cons. (↑) |
|---|---|---|---|
| ImagPose | 4.882 | 0.2385 | 0.7315 |
| MVDream | 8.669 | 0.2322 | 0.7286 |
| Ours | **3.697** | **0.2604** | **0.7920** |

Table 6: **Quantitative ablation study of different alignment loss.**

## A.4 COMPARISON WITH MORE BASELINES

We provide comparisons with additional potential baselines, such as ImagPose (Shen & Tang, 2024) and MV-Dream (Shi et al., 2023b). Compared with ImagPose, our method enables the customization model to generate multi-view consistent images via lightweight adaptation with only 100 samples, while Imagpose requires 85000 images with diverse viewpoints. Besides, Imagpose

| Method | MV Cons. (↓) | CLIP-T (↑) | ID Cons. (↑) |
|---|---|---|---|
| Full-Attn. | 5.433 | 0.2593 | 0.7855 |
| Parameter | 6.129 | 0.2602 | 0.7678 |
| Ours | **3.697** | **0.2604** | **0.7920** |

Table 7: **Quantitative ablation study of different alignment loss.**

lacks the ability to control rich semantics, such as background changes via prompts, while our method preserves the precise semantic control as validated in Table 6. For MVDream, it exhibits limited multiview consistency and semantic control, probably because it has difficulty generalizing to the human domain.

## A.5 DISCUSSION OF SMPL FITTING

SMPL may introduce errors during body fitting, for instance, the body shape may undergo slight changes, and features like hair may not be accurately modeled. However, this does not affect the multi-view customization in the second stage, since our goal is to provide an initial depth condition

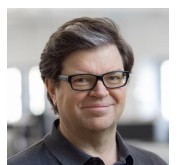
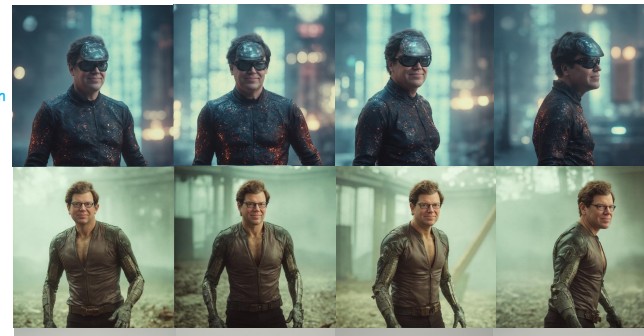

A futuristic man in a sleek cyberpunk outfit, neon lights reflecting on his chrome visor, dark sci-fi city background

A man with a robotic arm, leather jacket, standing in a post-apocalyptic ruin, intense gaze, gritty atmosphere

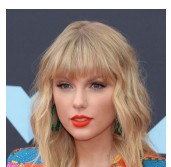

A woman wearing textured brown sweater and headphones

A futuristic woman in a silver exosuit floating in zero gravity, space station background, sci-fi fantasy style

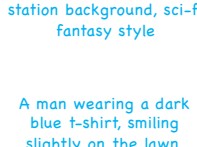

A man wearing a dark blue t-shirt, smiling slightly on the lawn. The expression on his face appears relaxed.

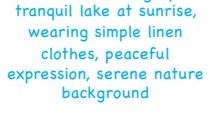

A man meditating by a tranquil lake at sunrise, wearing simple linen clothes, peaceful expression, serene nature background

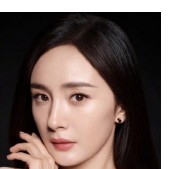

A woman wearing sunglasses and necklace, close-up

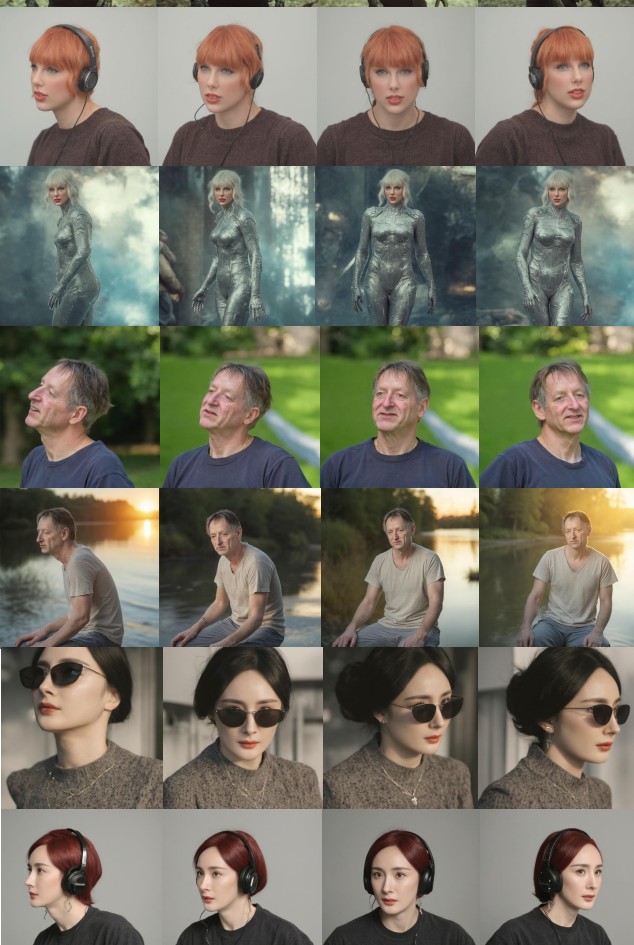

A woman wearing headphones with red hair

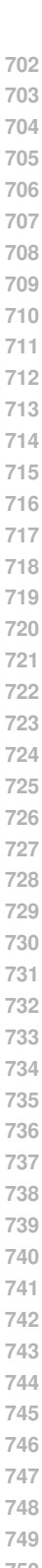

Figure 6: **More results of PersonalView.**

to ensure robust multi-view consistency and reasonable semantic alignment and it is not necessary for the body shape in Step 2 to precisely match the single-view image in Step 1. For example, we can control features like hair through prompts, as shown in Figure 4 and more results in Figure 6. In practice, our method demonstrates excellent multi-view consistency and prompt adherence. On the other hand, it does not introduce excessive inference costs, with a total GPU usage of 9.8GB and a runtime of 0.45s.

## A.6 DIFFERENCE FROM IC-LoRA.

In practice, IC-LoRA (Huang et al., 2024) boils down to fine-tuning the DiT on concatenated images, which is a simple yet widely-applicable technique. However, to our knowledge, there is no multi-view conditional generation model built on IC-LoRA at the time of submission. Our work provides solid

evidence of the success of in-context learning in transferring multi-view generation to a generic model. On the other hand, our experiments demonstrate the insufficiency of using in-context learning alone, which has distinct contributions not covered by IC-LoRA.

### A.7 LIMITATION

While our approach demonstrates strong performance in generating multi-view consistent customized images, it also has certain limitations. In particular, it primarily relies on activating the in-context learning capabilities of the base model, and therefore, its effectiveness is inherently constrained by the representational power of the underlying pre-trained model. Nevertheless, a key advantage of our framework lies in its modularity and transferability, making it easily adaptable to more powerful backbone models as they emerge — a direction we plan to explore in future work.

### A.8 ETHICS STATEMENT

Our main objective in this work is to empower novice users to generate visual content creatively and flexibly. However, we acknowledge the potential for misuse in creating fake or harmful content with our technology. Therefore, we believe it's essential to develop and implement tools to detect biases and malicious use cases to promote safe and equitable usage.

### A.9 REPRODUCIBILITY STATEMENT

We make the following efforts to ensure the reproducibility of PersonalView: (1) Our training and inference codes together with the trained model weights will be publicly available. (2) We provide training details in the appendix, which is easy to follow. (3) We provide the details of the human evaluation setups.

### A.10 LLM USAGE STATEMENT

Large Language Models (LLMs), specifically OpenAI's GPT-5, were employed as a general-purpose assistive tool during the preparation of this paper. The model was primarily used for: (1) Language polishing – refining grammar, improving clarity, and adjusting tone to meet academic writing standards. (2) Formatting support – generating LaTeX table templates, figure captions, and consistent section structuring. All core research activities—including problem formulation, theoretical development, model design, experiments, analysis, and conclusions—were entirely conceived and executed by the authors. The LLM was not used for generating original research ideas, deriving results, or writing substantive scientific content.

