# OpenReview forum: "PersonalView: Multi-View Consistent Human Image Customization via In-Context Learning"
_ICLR.cc/2026/Conference — ICLR 2026 Conference Withdrawn Submission_

### Official Review · Reviewer_NyfW · 2025-10-29

**Soundness:** 3
**Presentation:** 3
**Contribution:** 2
**Rating:** 2
**Confidence:** 4

**Summary:**

In this paper, they propose PersonalView, a text-to-image personalization method that enables consistent multi-view generation. To train an in-context LoRA for multi-view generation, they first construct multi-view depth maps using SMPL. During the training of the in-context LoRA on the obtained panel images, they introduce a Semantic Correspondence Alignment loss to prevent overfitting. This loss is applied to the attention maps between the pre-trained and fine-tuned models at each layer. To validate their approach, they conduct experiments demonstrating that PersonalView achieves superior multi-view consistency, identity preservation, and text alignment compared to various baselines, further supported by human evaluation results.

**Strengths:**

**S1**. The paper is well-structured and clearly presented, with a solid experimental evaluation. The paper is well-organized and easy to follow.

**S2**. The human evaluation and extensive ablation studies are thorough and well-executed, providing strong support for the proposed method.

**Weaknesses:**

**W1**. Limited novelty. The proposed method is too straightforward. In-context fine-tuning approaches (e.g., IC-LoRA) are already widely used, and the paper primarily introduces a new task of multi-view personalization along with a straightforward solution tailored to it, rather than presenting a fundamentally novel methodology. Moreover, the regularization loss proposed in the paper is also based on commonly used techniques, further limiting the methodological originality.

**W2**. The motivation for scenarios requiring consistent multi-view image generation is not clearly convincing. It would be helpful to provide a more concrete justification, for instance by illustrating its potential utility for 3D reconstruction or other practical downstream applications.

**W3**. In recent research, human personalization tasks have been extended to handle multiple identities within a single generated image. While some existing personalization modules already support multi-human scenarios, it remains unclear whether the proposed method can generalize to such tasks.

**Questions:**

**Q1**. Since proposed method is based on SMPL, it seems that the method could offer better controllability during inference compared to approaches using OpenPose. Are there any use cases or experiments demonstrating this advantage? Highlighting such scenarios could make the contribution more compelling.

**Q2**. In the qualitative examples, most subjects still appear to face the front or are shown in relatively simple backgrounds. How does the method perform in more challenging or extreme cases, such as when the person appears smaller in the scene or is viewed from nearly the back? Does it also maintain background consistency in such scenarios?

**Details Of Ethics Concerns:**

No concern.

---

### Official Review · Reviewer_Kt3F · 2025-10-31

**Soundness:** 3
**Presentation:** 3
**Contribution:** 2
**Rating:** 4
**Confidence:** 5

**Summary:**

This work addresses the limitations of existing personalized image generation methods, which often fail to control viewpoint and produce geometrically and visually consistent multi-view outputs. To this end, the authors propose PersonalView, a lightweight adaptation framework that enables pre-trained diffusion models to generate multi-view consistent human images using as few as 100 training samples. PersonalView leverages in-context learning by conditioning on a grid of multi-view depth maps—generated via 3D body fitting—as spatial priors to enforce cross-view coherence. To preserve prompt controllability and identity fidelity during fine-tuning, the method introduces a Semantic Correspondence Alignment Loss that aligns cross-modal attention between a frozen pretrained model and the trainable branch. Extensive evaluations show that PersonalView significantly outperforms baselines trained on large multi-view datasets, achieving superior performance in multi-view consistency, identity preservation, and text alignment.

**Strengths:**

1. The overall writing is easy to follow.
2. A novel task Multi-View Customization (MVC) is proposed, which aims to generate identity consistent multi-view customized images given single reference image.
3. A grid-based sampling strategy is proposed to enhance the correspondence between multi-view images. This leverages known priors in powerful foundation models without requiring architectural changes.
4. The proposed tuning strategy is highly data-efficient and practical compared to full-model fine-tuning or training from scratch.

**Weaknesses:**

1. Lack of Novelty and Conceptual Insight: The proposed method appears more as an engineering pipeline than a conceptually novel contribution. It combines existing techniques—depth-conditioned control (e.g., ControlNet-style conditioning) and in-context learning via grid prompting (e.g., IC-LoRA)—into a serial workflow without introducing a fundamental advance in modeling or understanding.
2. High Computational Cost with Scalability Issues: The grid-based in-context generation introduces non-trivial auxiliary computational overhead. Since the method arranges all target views into a grid-liked unified image, the computational cost grows quadratically with the number of desired output views.
3. Insufficient Identity Consistency in Challenging Views: Despite claims of strong identity preservation, the visual results show noticeable degradation in identity fidelity under large viewpoint variations. In several cases (Figure 4), especially with extreme angles, the generated faces and body proportions appear distorted or unnatural, indicating that the method struggles to maintain coherent identity across all poses.

**Questions:**

Please refer to the weakness.

---

### Official Review · Reviewer_Nmok · 2025-11-01

**Soundness:** 2
**Presentation:** 2
**Contribution:** 2
**Rating:** 4
**Confidence:** 4

**Summary:**

This paper introduces "Multi-View Customization" (MVC), a new task for generating multi-view consistent, personalized human images from a single reference. The proposed method, PersonalView, adapts a pre-trained Diffusion Transformer (FLUX) using in-context learning. It conditions the model on a grid of multi-view depth maps (derived from an SMPL model fit to an initial generation) to enforce geometric consistency. To preserve prompt controllability and identity fidelity during this lightweight adaptation (on only 100 samples), the authors introduce a novel Semantic Correspondence Alignment (SCA) loss, which aligns attention maps between the fine-tuned adapter and the frozen pre-trained model.

**Strengths:**

- The task of Multi-View Customization (MVC) is well-motivated and targets a meaningful problem with clear practical application scenarios.

- Effecient and Practical Solution: The core idea of using in-context learning with depth maps to activate a DiT's multi-view priors is a practical approach. And using only 100 training samples is efficient.

- The ablations provide clear evidence for the necessity and efficacy of both the multi-view in-context learning (MV-IL) and the SCA loss. And the method significantly outperforms the reported baselines in multi-view consistency, identity preservation, and text alignment.

**Weaknesses:**

1. Incremental Methodological Contribution: The proposed method is primarily a practical integration of existing techniques. The use of in-context learning for diffusion transformers, conditioning on 3D priors (like depth), and using attention-based losses for semantic preservation are all established patterns. The main contribution is the specific assembly for this task and the SCA loss, which, while useful, feels more like a solid engineering solution than a fundamental methodological advance.

2. Inadequate Baseline Comparison: The experimental comparison is not fully convincing.

- Unfair Prompting: Instruction-based models like Qwen-Image are not tested fairly. To properly evaluate their "multi-view customization" ability, they should be prompted with explicit instructions, e.g., "generate a [left view / 30-degree view], a [front view], and a [right view / -30-degree view] of the same person in the same scene...". The current, simpler prompts likely do not trigger their full instruction-following potential for this task.

- Missing Baselines: Key relevant baselines are missing, such as FLUX-Kontext, which is specifically designed for in-context generation with DiTs and would be a more direct comparison for the in-context learning component.

3. Incomplete Related Work: The literature review overlooks several key works, limiting the paper's contextualization. For example, "ICEdit" [1] is highly relevant to the discussion of in-context learning for instructional editing in DiTs. On the personalization front, "PhotoVerse" [2] presents a tuning-free customization approach that would be important to contrast with.

4. Dependency on SMPL and Lack of Generalizability: The method's reliance on SMPL for the depth scaffold is a significant weakness. This dependency strictly limits the method to human personalization and demonstrates a lack of generalizability, making it unsuitable for multi-view customization of arbitrary objects.

[1] In-Context Edit: Enabling Instructional Image Editing with In-Context Generation in Large Scale Diffusion Transformer
[2] PhotoVerse: Tuning-Free Image Customization with Text-to-Image Diffusion Models

**Questions:**

1. Following on Weakness #2: Could the authors provide detailed inputs including prompts for all baselines , and provide additional results using more explicit, view-specific prompts as suggested? This would be necessary for a fair comparison.

2. Following on Weakness #4: How does the model learn to synthesize complex, prompt-specifit geometry (like a "cyberpunk outfit" or "superman costume") when the conditioning depth map is just a simple, unclothed body model? Does this not limit the geometric variety of the generated clothing?

---

### Official Review · Reviewer_N6th · 2025-11-01

**Soundness:** 3
**Presentation:** 2
**Contribution:** 2
**Rating:** 4
**Confidence:** 3

**Summary:**

PersonalView activates the multi-view prior of DiT with the in-context condition of the multi-view depth grid, and constrains the text-image semantic alignment in fine-tuning with SCA to achieve multi-view consistent portrait customization with identity fidelity and controllable prompts.

**Strengths:**

The problem of being unable to control the perspective/poor consistency among multiple views is very interesting. It can simultaneously enhance geometric/visual consistency while maintaining controllable prompts and identity consistency. Under multiple indicators, SOTA results were achieved using a small sample size.

**Weaknesses:**

I would like to ask whether this method relies on SMPL fitting and depth quality, and is insufficient in quantifying the robustness and failure rate of occlusion errors. When there is a mismatch or strong occlusion in SMPL depth, can SCA still stably maintain semantic controllability and identity fidelity?

**Questions:**

I want to know what the outcome will be in more complex scenarios and interactions?

---

### Note · Authors · 2025-11-13

I have read and agree with the venue's withdrawal policy on behalf of myself and my co-authors.